# Quantifying Cognitive Function in Diabetes: Relationships Between AD8 Scores, HbA1c Levels, and Other Diabetic Comorbidities

**DOI:** 10.3390/biomedicines13020340

**Published:** 2025-02-03

**Authors:** Hsin-Yu Chao, Ming-Chieh Lin, Tzu-Jung Fang, Man-Chia Hsu, Ching-Chao Liang, Mei-Yueh Lee

**Affiliations:** 1Division of Endocrinology and Metabolism, Department of Internal Medicine, Kaohsiung Medical University Hospital, Kaohsiung 807378, Taiwan; c0928158920@gmail.com (H.-Y.C.); u102001122@gap.kmu.edu.tw (M.-C.L.); tzujung66@gmail.com (T.-J.F.); 2Division of Geriatrics and Gerontology, Department of Internal Medicine, Kaohsiung Medical University Hospital, Kaohsiung 807378, Taiwan; 3School of Medicine, College of Medicine, Kaohsiung Medical University, Kaohsiung 807378, Taiwan; 4Department of Nursing, Kaohsiung Medical University Gangshan Hospital, Kaohsiung 820111, Taiwan; k3338359@gmail.com; 5Department of Laboratory Medicine, Kaohsiung Municipal Siaogang Hospital, Kaohsiung 812015, Taiwan; k670806@yahoo.com.tw; 6Department of Internal Medicine, Kaohsiung Medical University Gangshan Hospital, Kaohsiung 820111, Taiwan

**Keywords:** Ascertain Dementia 8 questionnaire, type 2 diabetes mellitus, cognitive dysfunction, estimated glomerular filtration rate, peripheral artery disease

## Abstract

Background/Objectives: Dementia associated with diabetes mellitus (DM) has been well documented in the literature, but studies utilizing early screening tools to target populations with mild cognitive dysfunction remain limited. This study aimed to investigate early cognitive decline by studying the relationships between “Ascertain Dementia 8” (AD8) questionnaire scores and glycemic control, lipid profiles, estimated glomerular filtration rate (eGFR), and the complications of diabetes. Methods: This case–control, cross-sectional, observational study was conducted at a medical center and an affiliated regional hospital in southern Taiwan from 30 June 2021 to 30 June 2023. Patients diagnosed with type 2 diabetes mellitus aged ≥40 years were recruited. Their past medical history, biochemical data, and AD8 score were collected at the same time. Results: The patients with glycated hemoglobin (HbA1c) levels of ≥7% had a higher risk of cognitive impairment than those with HbA1c levels of <7% (*p* < 0.001). The participants whose eGFR was <60 mL/min/1.73 m^2^ had a higher mean AD8 score compared to those with an eGFR of ≥60 mL/min/1.73 m^2^ (*p* = 0.008). The patients with a medical history of peripheral artery disease and diabetic neuropathy were also associated with a higher mean AD8 score (*p* < 0.001 and *p* = 0.017, respectively). Conclusions: By employing the AD8 questionnaire as a sensitive screening tool, our study suggests that early cognitive decline is significantly associated with poorer glycemic control, a lower glomerular filtration rate, peripheral artery disease, and diabetic neuropathy. Early detection of these risk factors may facilitate timely interventions and tailored treatment strategies to treat or prevent cognitive dysfunction.

## 1. Introduction

Dementia is a significant cognitive disorder that greatly impacts quality of life, yet developing effective treatments to prevent or reverse its progression remains difficult [1]. In recent decades, the focus of treatment strategies has shifted to treating mild cognitive impairment (MCI) [2], which serves as a transitional stage between normal cognitive abilities and dementia. During this phase, individuals experience cognitive issues and measurable deficits in cognitive assessments while maintaining daily functioning [3]. MCI is primarily diagnosed through clinical evaluation, but the varying criteria and processes used globally lead to inconsistencies in diagnosis and subsequent investigations [4]. While the Mini-Mental Status Exam (MMSE) is not a definitive diagnostic tool for MCI, it can help identify individuals who may need further evaluation. A score of 19–23 suggests mild cognitive impairment [5].

The “Ascertain Dementia 8” questionnaire (AD8) is a common tool used by neurologists around the world to identify and detect early dementia or MCI [6]. The questionnaire is a user-friendly screening tool with eight questions that quickly evaluates memory, orientation, and judgment [7]. With a cut-off of two endorsed items, the questionnaire has a sensitivity of 72 to 91% and a specificity of 67 to 78% for identifying MCI [8,9]. The Chinese version of the AD8 questionnaire also shows a high sensitivity of 95.9% and specificity of 78.1% for MCI [10]. An AD8 score of 2 or greater strongly suggests cognitive impairment [9] and is effective for assessing cognitive statuses in community, primary care, and emergency settings [11].

Type 2 diabetes mellitus (T2DM) has emerged as a major risk factor for cognitive impairment and the development of dementia, as extensive research has demonstrated its significant association with an increased risk of cognitive decline and neurodegenerative disorders [12]. Both T2DM and dementia are rapidly increasing in prevalence due to factors such as population aging and lifestyle changes, leading to significant future healthcare and societal burdens. Thus, the identification of risk factors, along with effective prevention and treatment methods, has become increasingly important [13].

The association between T2DM and cognitive decline is multifaceted, involving alterations in brain structure and function that can lead to conditions such as Alzheimer’s disease (AD) and vascular dementia (VD) [14,15]. The affected functions include episodic memory and executive function [13]. Notably, patients with T2DM exhibit brain atrophy and microvascular disease, highlighting the intricate relationship between diabetes and cognitive health [14].

The mechanisms connecting T2DM and cognitive dysfunction include the following: (1) the abnormal glucose metabolism and hyperglycemia produce advanced glycation end products (AGEs), which lead to endothelial damage, inflammation, and oxidative stress; (2) factors promoting insulin resistance, such as dyslipidemia and obesity, disrupt the crucial actions of insulin that are necessary for memory functions in the brain, and they also contribute to the production of intracellular neurofibrillary tangles and extracellular Aβ plaques; (3) more common vasculopathic consequences in T2DM, such as lacunes, contribute to abnormalities in the small cerebral perforating arterioles, including arteriolosclerosis, lipohyalinosis, or fibrinoid necrosis, which lead to an increased incidence of VD and increased progression risk for a pre-existing cognitive decline [13,14,16]. T2DM patients with manifestations of microvascular or macrovascular disease are also more likely to have worse cognitive performance and have an increased risk of dementia [14]. Additionally, lower estimated glomerular filtration rates (eGFRs) were found to correlate with a higher dementia incidence [17].

From a precision health perspective, the early detection of diabetic patients at high risk for developing complications can facilitate timely interventions and tailored treatment strategies, potentially improving outcomes and mitigating the progression of the associated conditions. Primary prevention of T2DM, kidney dysfunction, and other vascular risk factors is important for halting the rapid rise in cognitive disorders associated with aging. Secondary prevention to avoid further deterioration of cognitive function and reduce the impact of early-stage disease should also be emphasized [13]. A rapid and effective screening tool, along with the identification of relevant risk factors based on the mechanisms through which T2DM affects cognitive function, as discussed above, is crucial for the early monitoring and management of cognitive decline and its related conditions.

Although cognitive decline associated with T2DM is well documented [18], studies studying the relationships between AD8 scores and diabetes-specific complications remain limited. Most existing studies correlated T2DM complications or biochemical parameters with cognitive dysfunction by focusing on patients that had already been diagnosed with dementia; however, the importance of identifying at-risk individuals at earlier stages such as MCI, a critical stage for intervention before the onset of advanced dementia, needs to be emphasized. Therefore, our study investigated cognitive decline using the highly sensitive AD8 score as a screening tool, examining its relationship with glycemic control, vascular disease risk factors, estimated glomerular filtration rates (eGFRs), and diabetes-related complications. Our findings aim to provide a novel approach for the early identification of high-risk individuals.

## 2. Materials and Methods

This case–control, cross-sectional, observational study was conducted at a medical center and an affiliated regional hospital in southern Taiwan from 30 June 2021 to 30 June 2023. The study was approved by the ethics committee prior to participant enrollment (KMUHIRB-E(I)-20190115), and informed consent was obtained from all the participants.

We enrolled patients aged ≥40 years diagnosed with type 2 diabetes mellitus, who were followed at the endocrine clinic of the Department of Internal Medicine, Kaohsiung Medical University Chung-Ho Memorial Hospital. Each participant was assessed using the Chinese version of the AD8 questionnaire [10]. The data collected at the time of AD8 scoring included age, sex, body mass index (BMI), past medical history (including diabetic comorbidities), fasting glucose level, glycated hemoglobin (HbA1c) level, alanine transaminase (ALT) level, lipid profile, estimated glomerular filtration rate (eGFR), and urine albumin-to-creatinine ratio (UACR). The participants with incomplete clinical or biochemical data; a history of traumatic brain injury, seizure/epilepsy, or psychiatric disorders; or those taking medications affecting cognitive function were excluded from the study. By excluding these patients with well-established independent risk factors for cognitive impairment, we aimed to isolate the impact of diabetes-related factors on cognitive decline.

Descriptive statistical analyses were conducted, and the participants were categorized for comparison based on the following criteria [19,20]:

Sex (male, female);

BMI (≥27 kg/m^2^, <27 kg/m^2^);

Fasting glucose level (≥130 mg/dL, <130 mg/dL);

HbA1c level (≥7%, <7%);

Total cholesterol level (≥200 mg/dL, <200 mg/dL);

Triglyceride level (≥150 mg/dL, <150 mg/dL);

High-density lipoprotein cholesterol (HDL-C) level (≥50 mg/dL, <50 mg/dL);

Low-density lipoprotein cholesterol (LDL-C) level (≥100 mg/dL, <100 mg/dL);

eGFR (≥60 mL/min/1.73 m^2^, <60 mL/min/1.73 m^2^);

UACR (≥30 mg/g, <30 mg/g).

Mean AD8 scores were compared using the chi-square test for bivariate analyses. Additionally, independent-samples *t*-tests were used to compare mean AD8 scores between patients with and without a history of diabetic comorbidities (including hypertension, ischemic stroke, myocardial infarction, peripheral arterial disease (PAD), diabetic retinopathy, and diabetic neuropathy). Mean values and standard deviations were calculated for each group. Biochemical data were compared between the patients with AD8 scores of ≥2 and those with scores of <2 using independent *t*-tests. A power analysis was conducted to determine the minimum sample size required to detect meaningful differences in cognitive impairment across the subgroups. Using an effect size of 0.5 (Cohen’s d), a significance level of 0.05, and a power of 95%, we calculated the minimum sample size to be 176 participants (88 per group) for independent-samples *t*-tests. This analysis was performed using the G*Power software (version 3.1.9.7).

Odds ratios (ORs) and 95% confidence intervals (CIs) for an AD8 score of ≥2 were calculated for those with and without diabetic comorbidities, as well as for the categorized biochemical data. These odds ratios were multivariate-adjusted using logistic regression, accounting for factors such as sex, age, underlying comorbidities, and biochemical markers. The statistical analysis was performed using IBM SPSS version 19.0.

## 3. Results

### 3.1. Baseline Characteristics of Study Population

Initially, 1034 subjects were enrolled, and a total of 909 patients with type 2 diabetes mellitus aged ≥40 years were analyzed, after excluding those not meeting the inclusion criteria. The mean age of the participants was 72.76 ± 5.75 years, with 549 females (60.4%) and 360 males (39.6%). The mean BMI was 25.8 ± 4.16 kg/m^2^; 585 patients (64.4%) had a BMI of <27 kg/m^2^ and 324 patients (35.6%) had a BMI of ≥27 kg/m^2^.

The mean fasting glucose level was 126.42 ± 47.17 mg/dL, with 597 patients (65.7%) having a fasting glucose level of <130 mg/dL and 312 patients (34.3%) with a level of ≥130 mg/dL. The mean HbA1c level was 6.76 ± 1.02%, with 608 patients (66.9%) with a level of <7% and 301 patients (33.1%) with a level of ≥7%.

The lipid profiles showed a mean total cholesterol level of 160.16 ± 34.91 mg/dL, with 811 patients (89.2%) with a level of <200 mg/dL and 98 patients (10.8%) with a level of ≥200 mg/dL. The mean triglyceride level was 109.65 ± 64.68 mg/dL, with 723 patients (79.5%) with a level of <150 mg/dL and 186 patients (20.5%) with a level of ≥150 mg/dL. The mean HDL-C level was 98.70 ± 50.57 mg/dL, with 484 patients (53.2%) with a level of <50 mg/dL and 425 patients (46.8%) with a level of ≥50 mg/dL. The mean LDL-C level was 81.58 ± 27.24 mg/dL, with 729 patients (80.2%) with a level of <100 mg/dL and 180 patients (19.8%) with a level of ≥100 mg/dL.

In terms of kidney function, the mean eGFR was 72.55 ± 24.41 mL/min/1.73 m^2^, with 652 patients (71.7%) with an eGFR of ≥60 mL/min/1.73 m^2^ and 257 patients (28.3%) with an eGFR of <60 mL/min/1.73 m^2^. The mean UACR was 124.78 ± 530.92 mg/g, with 605 patients (66.6%) with a UACR of <30 mg/g and 304 patients (33.4%) with a UACR of ≥30 mg/g.

Regarding comorbidities, 645 participants (71.0%) had hypertension, 22 (2.4%) had ischemic stroke, 21 (2.3%) had myocardial infarction, 55 (6.1%) had PAD, 151 (16.6%) had diabetic retinopathy, and 153 (16.8%) had diabetic neuropathy. The detailed baseline characteristics are presented in Table 1.

### 3.2. Sex, BMI, and Mean AD8 Score

The mean AD8 score was significantly higher in females (0.83) than in males (0.59) (*p* = 0.001). The OR for an AD8 score of ≥2 in females compared to males was 1.749 (95% CI: 1.174 to 2.605). No significant difference in mean AD8 scores was observed between the patients with a BMI of ≥27 kg/m^2^ (0.75) and those with a BMI of <27 kg/m^2^ (0.72) (*p* = 0.487).

### 3.3. Glucose Profiles and Mean AD8 Score

The patients with HbA1c levels of ≥7% had a higher mean AD8 score (1.01) (*p* < 0.001) compared to those with HbA1c levels of <7% (0.60). The OR for an AD8 score of ≥2 in the patients with HbA1c levels of ≥7% compared to those with HbA1c levels of <7% was 2.33 (95% CI: 1.560 to 3.472). There was no significant difference in mean AD8 scores between the patients with fasting glucose levels of ≥130 mg/dL (0.76) and those with levels of <130 mg/dL (0.70) (*p* = 0.558).

### 3.4. Lipid Profiles and Mean AD8 Score

The individuals with total cholesterol levels of ≥200 mg/dL had a lower mean AD8 score (0.51) compared to those with levels of <200 mg/dL (0.76) (*p* = 0.066). Similarly, the individuals with LDL-C levels of ≥100 mg/dL had a mean AD8 score of 0.59, compared to 0.77 for those with LDL-C levels of <100 mg/dL (*p* = 0.113). The participants with triglyceride levels of ≥150 mg/dL showed a slightly higher mean AD8 score (0.83) compared to those with levels of <150 mg/dL (0.71) (*p* = 0.563). There were no significant differences in mean AD8 scores among the groups for the other lipid parameters.

### 3.5. Kidney Function and Mean AD8 Score

The participants with an eGFR of <60 mL/min/1.73 m^2^ had a significantly higher mean AD8 score (0.86) compared to those with an eGFR of ≥60 mL/min/1.73 m^2^ (0.68) (*p* = 0.008). No significant difference was found in mean AD8 scores between the patients with a UACR of ≥30 mg/g (0.81) and those with a UACR of <30 mg/g (0.70) (*p* = 0.297).

### 3.6. Other Macrovascular and Chronic Complications of Diabetes and Mean AD8 Score

The patients with a history of PAD had a significantly higher mean AD8 score (1.33) than those without (*p* < 0.001). Similarly, those with diabetic neuropathy had a higher mean AD8 score (0.95) compared to those without (*p* = 0.017). The OR for an AD8 score of ≥2 in the patients with PAD compared to those without was 1.943 (95% CI: 1.016 to 3.717). Increased mean AD8 scores were noted in the groups with hypertension (0.76), ischemic stroke (0.68), myocardial infarction (0.81), and diabetic retinopathy (0.74), but they did not reach statistical significance.

The differences in mean AD8 scores are presented in Table 2 and the ORs for an AD8 score of ≥2 and specific biochemical parameters and diabetic comorbidities are presented in Figure 1.

### 3.7. Comparison of Biochemical Data Between Patients with AD8 Scores of ≥2 and <2

A total of 156 participants (17.7%) had an AD8 score of ≥2. The mean age in this group (74.61 ± 6.79 years) was significantly higher than that of the AD8 score < 2 group (72.38 ± 5.44 years) (*p* < 0.001). The mean HbA1c level in the AD8 score ≥ 2 group (7.18 ± 1.11%) was also significantly higher than that of the AD8 score < 2 group (6.68 ± 0.98%) (*p* < 0.001). No significant differences were found in the BMI, fasting glucose level, lipid profiles, eGFR, or UACR between the two AD8 score groups. The biochemical differences between the AD8 score groups are presented in Table 3.

## 4. Discussion

### 4.1. Sex and AD8 Score

This study provides compelling evidence of the significant association between T2DM and cognitive decline, as indicated by the AD8 scores. The female participants had a significantly higher mean AD8 score compared to the male participants in this study, and the OR of an AD8 score of ≥2 in females was 1.749 (95% CI, 1.174 to 2.605). This finding is in agreement with the results of Cholerton B et al., who found that cognition function declines more rapidly among older female T2DM patients [13]. In a pooled analysis of 2.3 million people comprising more than 100,000 cases of dementia, the relative risk (RR) of any type of dementia for women with diabetes was 1.62 (95% CI: 1.45–1.80), while for men it was 1.58 (95% CI: 1.38–1.81) [21].

The observed sex difference in AD8 scores may be partially attributed to age-related changes in sex hormones. Estrogens are known to exert neuroprotective effects on the central nervous system (CNS) through both genomic and non-genomic mechanisms, regulating neurotransmitter activity, synaptic function, and neuroplasticity, while safeguarding against neurotoxic insults [22]. Studies suggest that the loss of estrogen might increase the risk of developing neurodegenerative diseases later in life [23,24], potentially explaining the higher AD8 scores of the female participants in our study. Additionally, women generally have a longer lifespan than men, which may increase their vulnerability to age-related cognitive decline. Women have also been reported to be twice as likely as men to experience depression, particularly during menopause, which may further increase their risk of Alzheimer’s disease [25]. While these mechanisms are supported by the existing literature, further studies are needed to confirm their relevance to our findings and to explore additional factors that may contribute to this observed difference.

### 4.2. Glucose Profiles and AD8 Score

Among the 909 patients analyzed in our study, 17.7% demonstrated cognitive impairment with an AD8 score of 2 or greater. A significantly higher HbA1c level (7.18 ± 1.11%) was observed in the AD8 score ≥ 2 group, which is highly indicative of potential cognitive impairment. The OR of 2.33 for an AD8 score of ≥2 suggests that patients with HbA1c levels of ≥7% are more likely to have cognitive impairment compared to those with HbA1c levels of <7%, highlighting a clear link between poor glycemic control and cognitive dysfunction.

Multiple studies have investigated the association between glucose metabolism and the risk of dementia. The ACCORD-MIND trial demonstrated a relationship between baseline glycemic control and cognitive function in individuals with type 2 diabetes, highlighting that poor glycemic control may be associated with cognitive decline [26]. Similar findings were noted in large Western cohort studies [12,27,28] and a secondary analysis study [29]. Our findings are consistent with those of multiple studies that highlighted the detrimental effects of hyperglycemia on brain health, suggesting that chronically elevated glucose levels may lead to cognitive impairment.

The relationship between blood glucose levels and cognitive impairment has been extensively explored in the literature. Persistent high blood glucose levels can induce various pathophysiological changes, including oxidative stress, inflammatory responses, and microvascular complications, which may lead to lipid peroxidation, protein misfolding, and deoxyribonucleic acid (DNA) damage, all of which can compromise neuronal function [13,16]. Notably, in individuals with diabetes, these changes are recognized as significant contributors to cognitive impairment. Studies have suggested that the effects of chronic hyperglycemia extend beyond peripheral glucose regulation and involve the role of insulin in the central nervous system. The neuroprotective effects of insulin in the brain are compromised, potentially increasing the risk of Alzheimer’s disease and other forms of cognitive impairment. The lack of insulin in the hippocampus can lead to neuronal death and decreased neuroplasticity, adversely affecting learning and memory functions [14]. A recent review demonstrated that sodium–glucose cotransporter 2 inhibitors (SGLT2is) have the ability to reduce the risk of dementia by remodeling glucotoxicity or reducing hyperphosphorylated tau levels and amyloid β accumulation in the brain; however, other studies on this subject had inconclusive results [30]. Neuroimaging studies have also revealed significant structural changes in the hippocampus and cortical areas of individuals with diabetes due to brain volume reductions and white matter degeneration, which correlated closely with declines in cognitive function. Moreover, microvascular damage may result in inadequate blood flow to the brain, exacerbating neuronal injuries [31,32,33]. These biological mechanisms explain our results that showed that higher HbA1c levels correlated with more cognitive impairment.

### 4.3. Kidney Function, UACR, and AD8 Score

This study identified a significant relationship between kidney function, as measured by the eGFR, and cognitive performance. The patients with an eGFR below 60 mL/min/1.73 m^2^ had a significantly higher mean AD8 score (0.86) than those with better kidney function (0.68). This result is supported by a large registry-based study in Stockholm, which demonstrated that lower eGFRs and steeper kidney function declines were associated with the development of dementia in the residents aged ≥65 years [17]. In another meta-analysis in 468,699 Scandinavians, including the Copenhagen general population, the random-effects risk of dementia was 1.14 (1.06–1.22) for a mildly decreased eGFR (60–90 mL/min/1.73 m^2^), 1.31 (0.92–1.87) for a moderately decreased eGFR (30–59 mL/min/1.73 m^2^), and 1.91 (1.21–3.01) for a severely decreased eGFR (<30 mL/min/1.73 m^2^), compared to the reference eGFR (>90 mL/min/1.73 m^2^) [34].

The influence of renal dysfunction on cognitive impairment manifests across various domains. Berger et al. demonstrated that patients with chronic kidney disease (CKD) exhibit significantly lower cognitive function compared to healthy controls, particularly in executive function and attention [35]. The metabolic disturbances and accumulation of endogenous toxins, such as uremic toxins, resulting from chronic kidney disease can directly harm the nervous system. Tang et al. noted that these toxins could induce neuroinflammation and oxidative stress, adversely affecting neuronal function and survival [36]. Prior research has established that the elevated homocysteine levels and uremia associated with CKD are linked to the presence of white matter lesions and the development of Alzheimer’s disease [37,38]. These associations are thought to arise from both direct prothrombotic effects and inflammatory responses within the endothelium [39,40]. Elevated cystatin-C levels in patients with CKD have also been correlated with an increased risk of developing Alzheimer’s disease [41]. In addition, other metabolic toxins that accumulate due to renal impairment—such as phosphate and fibroblast growth factor 23 [42]—along with certain guanidine compounds, like creatinine and guanidinosuccinic acid, may further exacerbate declines in brain function [43]. The implications of these findings emphasize the intricate interplay between metabolic health, kidney function, and cognitive outcomes, suggesting that renal impairment may contribute to or exacerbate the cognitive decline in T2DM patients.

In our study, a UACR ≥30 with a higher mean AD8 score, although it did not reach statistical significance, may indicate early microvascular damage, potentially contributing to neuroinflammation and endothelial dysfunction. Albuminuria has been identified as an independent risk factor for cognitive impairment and dementia in recent systematic reviews and cohort studies [44,45,46]. Additionally, urine albumin has been suggested as a potential early marker for CKD and cognitive dysfunction. The mechanisms underlying this association are likely multifactorial. The shared microvascular pathogenesis in the kidneys and brain may explain the link between albuminuria and cognitive impairment [47,48]. However, inconsistencies with previous findings could stem from differences in the study populations, the severity of albuminuria, or measurement methods [49]. For instance, our study’s single-timepoint UACR measurement may have diluted its association with cognitive outcomes due to variability from external factors [49]. Future studies with repeated UACR measurements and larger, diverse populations are needed to clarify these relationships.

### 4.4. BMI, Lipid Profiles, and AD8 Score

In this study, the participants with a BMI of ≥27 kg/m^2^ demonstrated higher mean AD8 scores, but the difference did not reach statistical significance. Similar findings were observed for the lipid profiles: the individuals with total cholesterol levels of ≥200 mg/dL and LDL-C levels of ≥100 mg/dL showed slightly lower mean AD8 scores, while those with triglyceride levels of ≥150 mg/dL and HDL-C levels of <50 mg/dL exhibited slightly higher mean AD8 scores. Nonetheless, none of these differences were statistically significant.

The literature on the correlation between BMI and cognitive function contains conflicting results. Unlike studies supporting the “obesity paradox”, where a higher BMI may protect against cognitive decline [50], our findings align with the evidence linking a higher BMI to poorer cognitive outcomes. Recent research has suggested that elevated BMI may contribute to systemic inflammation, insulin resistance, vascular dysfunction, and even changes in the macrophage phenotype within adipose tissue, all of which are associated with cognitive impairment [15]. However, Mendelian randomization studies, such as those from the United Kingdom (UK) Biobank, have found limited evidence for a causal relationship between BMI and cognitive function, highlighting the potential role of confounders or reverse causation [8,9,10,11,12,13,14,15,16,17,18,19,20,21,22,23,24,25,26,27,28,29,30,31,32,33,34,35,36,37,38,39]. For example, weight loss may reflect early neurodegenerative changes rather than a protective effect of adiposity [51]. These inconsistencies emphasize the need for longitudinal studies to clarify the relationship between BMI, metabolic health, and cognitive outcomes across different populations.

Regarding the lipid profiles, in the Fremantle Diabetes Study, elevated total cholesterol levels were associated with a reduced risk of cognitive impairment over eight years [52]. Conversely, other studies have linked elevated plasma triglyceride and cholesterol levels in individuals with type 2 diabetes to poorer cognitive function [53,54]. The ACCORD-MIND trial found no significant differences in cognitive decline between intensive and standard lipid-lowering therapies [55]. While high adiposity in midlife is strongly associated with cognitive decline and dementia, conclusions on the impact of late-life adiposity on short-term dementia risk remain inconsistent [56,57,58]. Consequently, the impact of dyslipidemia on cognitive impairment in T2DM remains uncertain. The lack of significant associations in our study may reflect the well-controlled lipid profiles in our participants due to routine clinical management. Differences in study design, cognitive assessment tools, and population characteristics could also account for the discrepancies with previous findings.

### 4.5. Diabetic Comorbidities or Complications and AD8 Score

The study also revealed significant associations between various macrovascular and chronic complications of diabetes and cognitive impairment in T2DM patients, as measured by the AD8 score. Diabetic neuropathy was associated with a significantly elevated mean AD8 score of 0.95. Furthermore, the patients with a history of PAD exhibited a markedly higher mean AD8 score (1.33) compared to those without PAD. The odds ratio of having an AD8 score of ≥2 in the patients with PAD was 1.943, indicating that these patients were more likely to exhibit cognitive impairment compared to those without PAD.

Currently, the evidence linking macrovascular or microvascular disease and cognitive impairment in diabetes is inconsistent and varies depending on the specific vascular area examined [59]. In a large, population-based, matched cohort study from Canada, the risk of dementia was greatest in those with the diabetic macrovascular complications of prior cerebrovascular disease (hazard ratio (HR): 2.03; 95% CI: 1.88–2.19) and PAD (HR: 1.47; 95% CI: 1.19–1.82) [60]. In the Fremantle Diabetes Study, PAD was found to be a strong independent risk factor (odds ratio: 5.35; 95% CI: 2.08–13.72) for dementia in diabetes patients [52]. Similar results demonstrated that type 2 diabetes patients with cognitive impairment were associated with a lower ankle–brachial index (ABI) [54,61]. The potential mechanisms underlying these findings include impaired oxygen supply and vascular deficits, which lead to chronic hypoxia, neuroinflammation, and mitochondrial dysfunction. These processes compromise neuronal survival and synaptic function, particularly in regions like the hippocampus, exacerbating cognitive decline [62]. We did not find significant correlations between cognitive impairment and certain complications, such as hypertension and retinopathy, and the evidence in the literature is inconsistent. Longitudinal studies suggest that early hypertensive episodes or retinal microvascular changes may be associated with cognitive decline, although the results are mixed and depend on the study population and methodology [63,64,65,66,67,68,69]. This reinforces the complex interactions among the various vascular risk factors and diabetic complications that influence cognitive outcomes.

### 4.6. Limitations and Future Directions

There are several limitations in this study. First, the cross-sectional design limits the ability to establish dynamic relationships between T2DM and cognitive decline. For example, it is unclear whether poor glycemic control directly contributes to cognitive decline or whether cognitive impairment indirectly impacts diabetes management. Single-timepoint measurements of glycemic control and cognitive performance may not capture long-term trends or fluctuations. Second, while we excluded patients with conditions such as traumatic brain injury and epilepsy to minimize confounding effects on cognitive outcomes, this decision may limit the generalizability of our findings to the broader T2DM population. These criteria were applied to isolate the impact of diabetes-related factors on cognitive impairment, which was the primary focus of our investigation. Third, the reliance on the AD8 questionnaire for cognitive assessment, while sensitive and user-friendliness in a clinical setting, does not provide a comprehensive evaluation of all cognitive domains. Additional neuropsychological tests such as the Montreal Cognitive Assessment (MoCA) or MMSE could enhance the accuracy of cognitive impairment detection and offer a more nuanced understanding of the specific cognitive deficits associated with T2DM. Fourth, our study did not account for certain potential confounders, such as medication use (e.g., antidiabetic or antihypertensive drugs), the duration of diabetes, and lifestyle factors (e.g., diet, physical activity, and smoking status). These factors may have contributed to the residual confounding effects in our analysis. Lastly, our sample was drawn from a single medical center and an affiliated regional hospital in southern Taiwan, which may limit the generalizability of the findings to other populations or geographic regions. Further multi-center, prospective, longitudinal studies using various cognitive tests may establish a more precise relationship between risk factors in diabetic patients and cognitive impairment.

## 5. Conclusions

This study demonstrated significant associations of poorer glycemic control, reduced kidney function, and diabetic complications (e.g., peripheral arterial disease and diabetic neuropathy) with higher AD8 scores, suggesting a potential link between these factors and early cognitive decline. The incorporation of cognitive health assessments, such as the AD8 questionnaire, into routine diabetes management protocols could be considered. The early identification of at-risk individuals may enable timely interventions and the development of tailored management strategies to mitigate cognitive dysfunction.

## Figures and Tables

**Figure 1 biomedicines-13-00340-f001:**
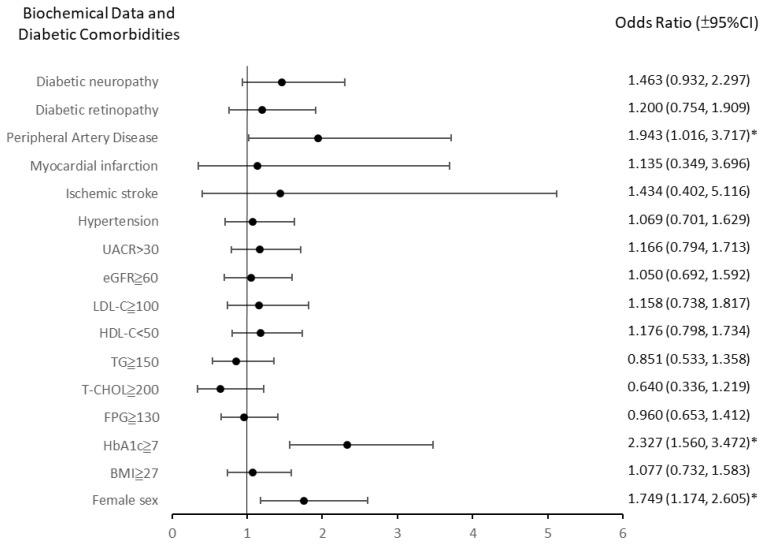
Odds ratios (ORs) and 95% confidence intervals (CIs) for patients with AD8 scores above 2 categorized by biochemical parameters and diabetic comorbidities. Variables with statistically significant associations are highlighted (*): peripheral artery disease (OR: 1.943; 95% CI: 1.016–3.717), HbA1c ≥ 7 (OR: 2.327; 95% CI: 1.560–3.472), and female sex (OR: 1.749; 95% CI: 1.174–2.605). Other variables, such as diabetic neuropathy and BMI ≥ 27, showed trends but were not statistically significant. AD8, “Ascertain Dementia 8” questionnaire; BMI, body mass index; HbA1c, glycated hemoglobin; FPG, fasting plasma glucose; T-CHOL, total cholesterol; HDL-C, high-density lipoprotein cholesterol; LDL-C, low-density lipoprotein cholesterol; eGFR, estimated glomerular filtration rate; UACR, urine albumin-to-creatinine ratio.

**Table 1 biomedicines-13-00340-t001:** Baseline characteristics of study population.

Characteristic	Total *n* (%) = 909 (100%)
Age (years), Mean ± SD	72.76 ± 5.75
Sex	
Male, *n* (%)	360 (39.6%)
Female, *n* (%)	549 (60.4%)
AD8 Score, Mean ± SD	0.73 ± 1.283
<2, *n* (%)	753 (82.3%)
≥2, *n* (%)	156 (17.7%)
BMI (kg/m^2^), Mean ± SD	25.8 ± 4.16
<27, *n* (%)	585 (64.4%)
≥27, *n* (%)	324 (35.6%)
HbA1c Level (%), Mean ± SD	6.76 ± 1.02
<7, *n* (%)	608 (66.9%)
≥7, *n* (%)	301 (33.1%)
Fasting Glucose Level (mg/dL), Mean ± SD	126.42 ± 47.17
<130, *n* (%)	312 (34.3%)
≥130, *n* (%)	597 (65.7%)
Total Cholesterol Level (mg/dL), Mean ± SD	160.16 ± 34.91
<200, *n* (%)	811 (89.2%)
≥200, *n* (%)	98 (10.8%)
Triglyceride Level (mg/dL), Mean ± SD	109.65 ± 64.68
<150, *n* (%)	723 (79.5%)
≥150, *n* (%)	186 (20.5%)
HDL-C Level (mg/dL), Mean ± SD	98.70 ± 50.57
<50, *n* (%)	484 (53.2%)
≥50, *n* (%)	425 (46.8%)
LDL-C Level (mg/dL), Mean ± SD	81.58 ± 27.24
<100, *n* (%)	729 (80.2%)
≥100, *n* (%)	180 (19.8%)
eGFR (mL/min/1.73 m^2^), Mean ± SD	72.55 ± 24.41
<60, *n* (%)	257 (28.3%)
≥60, *n* (%)	652 (71.7%)
UACR (mg/g), Mean ± SD	124.78 ± 530.92
<30, *n* (%)	605 (66.6%)
≥30, *n* (%)	304 (33.4%)
Diabetic Comorbidities	
Hypertension, *n* (%)	645 (71.0%)
Ischemic Stroke, *n* (%)	22 (2.4%)
Myocardial Infarction, *n* (%)	21 (2.3%)
Peripheral Arterial Disease, *n* (%)	55 (6.1%)
Retinopathy, *n* (%)	151 (16.6%)
Neuropathy, *n* (%)	153 (16.8%)

Data are presented as mean ± standard deviation (SD) for continuous variables and n (%) for categorical variables. AD8, “Ascertain Dementia 8” questionnaire; BMI, body mass index; HbA1c, glycated hemoglobin; HDL-C, high-density lipoprotein cholesterol; LDL-C, low-density lipoprotein cholesterol; eGFR, estimated glomerular filtration rate; UACR, urine albumin-to-creatinine ratio.

**Table 2 biomedicines-13-00340-t002:** Comparison of mean AD8 scores between groups.

Characteristic	Mean AD8 Score ± SD	*p*-Value *
Sex		
Male	0.59 ± 1.13	**0.001 ***
Female	0.83 ± 1.37	
BMI (kg/m^2^)		
<27	0.72 ± 1.31	0.487
≥27	0.75 ± 1.24	
HbA1c Level (%)		
<7	0.60 ± 1.10	**<0.001 ***
≥7	1.01 ± 1.55	
Fasting Glucose Level (mg/dL)		
<130	0.70 ± 1.27	0.558
≥130	0.76 ± 1.31	
Total Cholesterol Level (mg/dL)		
<200	0.75 ± 1.31	0.066
≥200	0.51 ± 0.97	
Triglyceride (mg/dL)		
<150	0.71 ± 1.26	0.563
≥150	0.83 ± 1.38	
HDL-C Level (mg/dL)		
≥50	0.71 ± 1.23	0.422
<50	0.76 ± 1.33	
LDL-C Level (mg/dL)		
<100	0.77 ± 1.34	0.113
≥100	0.59 ± 1.03	
eGFR (mL/min/1.73 m^2^)		
≥60	0.68 ± 1.18	**0.008 ***
<60	0.86 ± 1.51	
UACR (mg/g)		
<30	0.70 ± 1.25	0.297
≥30	0.81 ± 1.35	
Hypertension		
with	0.76 ± 1.27	0.817
without	0.67 ± 1.32	
Ischemic Stroke		
with	0.86 ± 1.04	0.284
without	0.73 ± 1.29	
Myocardial Infarction		
with	0.81 ± 1.25	0.945
without	0.73 ± 1.29	
Peripheral Arterial Disease		
with	1.33 ± 1.98	**<0.001 ***
without	0.70 ± 1.22	
Retinopathy		
with	0.74 ± 1.12	0.481
without	0.73 ± 1.32	
Neuropathy		
with	0.95 ± 1.52	**0.017 ***
without	0.69 ± 1.23	

The mean ± standard deviation (SD) of the AD8 scores of the categorized groups are presented. Bold *p*-values are < 0.05 (*) and were considered statistically significant. AD8, “Ascertain Dementia 8” questionnaire; BMI, body mass index; HbA1c, glycated hemoglobin; HDL-C, high-density lipoprotein cholesterol; LDL-C, low-density lipoprotein cholesterol; eGFR, estimated glomerular filtration rate; UACR, urine albumin-to-creatinine ratio.

**Table 3 biomedicines-13-00340-t003:** Comparison of biochemical data between groups with AD8 scores of ≥2 and <2.

	AD8 Score ≥ 2 (*n* = 156)	AD8 Score < 2 (*n* = 753)	*p*-Value *
Age (years)	74.61 ± 6.79	72.38 ± 5.44	**<0.001 ***
BMI (kg/m^2^)	26.19 ± 5.11	25.83 ± 3.94	0.326
Fasting Glucose Level (mg/dL)	126.45 ± 37.10	126.27 ± 49.03	0.839
HbA1c Level (%)	7.18 ± 1.11	6.68 ± 0.98	**<0.001 ***
Total Cholesterol Level (mg/dL)	160.31 ± 30.39	160.13 ± 35.80	0.951
Triglyceride Level (mg/dL)	110.46 ± 66.67	105.74 ± 54.13	0.407
HDL-C Level (mg/dL)	49.63 ± 13.20	50.77 ± 14.20	0.357
LDL-C Level (mg/dL)	82.67 ± 26.13	81.36 ± 27.49	0.586
eGFR (mL/min/1.73 m^2^)	70.66 ± 25.21	72.94 ± 24.24	0.236
UACR (mg/g)	154.39 ± 409.58	118.64 ± 552.81	0.349

Mean ± standard deviation (SD) values for biochemical and demographic variables between the two AD8 score groups (AD8 score ≥ 2 and AD8 score < 2) were compared. Bold *p*-values are <0.05 (*) and were considered statistically significant. AD8, “Ascertain Dementia 8” questionnaire; BMI, body mass index; HbA1c, glycated hemoglobin; HDL-C, high-density lipoprotein cholesterol; LDL-C, low-density lipoprotein cholesterol; eGFR, estimated glomerular filtration rate; UACR, urine albumin-to-creatinine ratio.

## Data Availability

The data presented in this study are available on request from the corresponding author. The data are not publicly available due to privacy restrictions.

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
