# Peer review of "Quantifying Cognitive Function in Diabetes: Relationships Between AD8 Scores, HbA1c Levels, and Other Diabetic Comorbidities"

_biomedicines, 2025, doi:10.3390/biomedicines13020340_

Round 1

Reviewer 1 Report (Previous Reviewer 3)

Comments and Suggestions for Authors

This study investigates the link between cognitive decline and factors like glycemic control, eGFR, and diabetes complications in type 2 diabetes patients. Results show that higher HbA1c levels, lower eGFR, and conditions like peripheral artery disease and diabetic neuropathy are associated with increased cognitive impairment. Early detection of these factors could enable timely intervention and tailored treatment for cognitive dysfunction in diabetes. While the study highlights important associations between diabetes-related factors and cognitive decline, it lacks novelty due to the presence of several similar studies in the field. Many existing studies already explore the connection between glycemic control, kidney function, and diabetes-related complications with cognitive impairment. The study does not introduce new mechanisms, therapeutic interventions, or innovative approaches to understanding or managing cognitive dysfunction in diabetes, making its findings less innovative compared to previous research. To enhance its novelty, the study could have explored novel biomarkers, intervention strategies, or longitudinal data to offer new insights into the relationship between diabetes and dementia.

Author Response

Comment:
This study investigates the link between cognitive decline and factors like glycemic control, eGFR, and diabetes complications in type 2 diabetes patients. Results show that higher HbA1c levels, lower eGFR, and conditions like peripheral artery disease and diabetic neuropathy are associated with increased cognitive impairment. Early detection of these factors could enable timely intervention and tailored treatment for cognitive dysfunction in diabetes. While the study highlights important associations between diabetes-related factors and cognitive decline, it lacks novelty due to the presence of several similar studies in the field. Many existing studies already explore the connection between glycemic control, kidney function, and diabetes-related complications with cognitive impairment. The study does not introduce new mechanisms, therapeutic interventions, or innovative approaches to understanding or managing cognitive dysfunction in diabetes, making its findings less innovative compared to previous research. To enhance its novelty, the study could have explored novel biomarkers, intervention strategies, or longitudinal data to offer new insights into the relationship between diabetes and dementia.

Reply:
We sincerely appreciate your valuable feedback and recognize the importance of introducing new mechanisms, therapeutic interventions, or longitudinal data to enhance the novelty of research in this field. While this study does not explore novel biomarkers or longitudinal designs, its primary contribution lies in the application of the AD8 questionnaire, which distinguishes our research from previous studies. While we acknowledge that many studies have explored the relationship between glycemic control, kidney function, and cognitive impairment, our study highlights the utility of the AD8 as a convenient and sensitive tool for screening cognitive decline in clinical and community settings. Unlike the more commonly used MMSE, which can be time-consuming and complex for routine use, the AD8 offers a practical alternative for clinicians, emphasizing early detection in real-world scenarios. [1-3]

Another important aspect of the AD8 is its ability to screen for possible mild cognitive impairment (MCI), a stage of cognitive decline that has not yet progressed to dementia. This is particularly significant in diabetes-related cognitive dysfunction, as MCI detection provides a critical window for timely intervention and tailored treatment strategies. While many studies focus on patients with clinically diagnosed dementia, our research addresses the gap by targeting an earlier stage of cognitive decline, which is often overlooked in the literature. We believe that this approach underscores the clinical relevance and applicability of our findings.

In addition to introducing the AD8 as an innovative tool, our study also simultaneously evaluated sex, BMI, multiple diabetes-related comorbidities (e.g., peripheral artery disease, diabetic neuropathy) and biochemical markers (e.g., HbA1c, eGFR, UACR, lipid profiles). This comprehensive approach allows us to capture a broader spectrum of factors contributing to cognitive dysfunction in diabetes, which has been less frequently addressed in existing literature. By examining the interplay between these variables, we provide clinicians with valuable insights into potential early risk factors for cognitive impairment in type 2 diabetes mellitus (T2DM) patients.

* References:

  1. J Clin Diagn Res. 2012 Dec;6(10):1658-62.
  2. Neurobiol Aging. 2008 Jul;29(7):1022-6.
  3. Br J Psychiatry. 2010 Jan;196(1):36-40.

To address the reviewer's concerns and to emphasize these key points, we have revised the manuscript to further highlight the uniqueness and contributions of our study. These revisions are marked in red for clarity and can be found in the “Abstract” on Page 1 (Lines 17-19, 31-32) and in the “Introduction” section on Page 1 (Lines 42-44, 46-50), Page 3 (Lines 100-107, 109-110).

Reviewer 2 Report (Previous Reviewer 2)

Comments and Suggestions for Authors

First of all, I want to thank all the authors for this nice piece of work

The abstract does not clearly define the unique contribution of the study. While it mentions the investigation of the relationship between diabetes-related comorbidities and cognitive decline, this area has been extensively studied. To better engage the reader, the abstract should explicitly highlight what makes this study distinct, such as specific methodologies, population characteristics, or novel insights. The introduction, although informative, fails to articulate the precise research gaps this study aims to address. Without this distinction, the study risks appearing as a reiteration of existing literature rather than a significant advancement.

The methods section contains critical issues regarding the inclusion and exclusion criteria. Excluding patients with traumatic brain injury and epilepsy is a logical step to minimize confounding variables. However, the rationale for these exclusions is insufficiently explained. This omission creates uncertainty about whether the sample is representative of the broader population of individuals with type 2 diabetes. Additionally, relying solely on the AD8 questionnaire for cognitive assessment introduces potential biases. Although AD8 is sensitive and user-friendly, it may not capture the full spectrum of cognitive impairments. Including a complementary cognitive evaluation would enhance the robustness and credibility of the findings. The description of the statistical methods is another area of concern. While power analysis is mentioned, the effect sizes and assumptions underlying this analysis are unclear. This lack of detail weakens confidence in the study's capacity to detect meaningful differences across subgroups.

The results section is well-organized but lacks deeper interpretive analysis. For instance, non-significant findings such as the relationship between lipid profiles and cognitive impairment are presented without sufficient discussion. This omission leaves readers with unanswered questions about potential explanations or the broader context of these findings. Variables like BMI and UACR show trends that, although not statistically significant, could hint at meaningful patterns. These trends should be elaborated upon, either by hypothesizing underlying reasons or proposing further investigation. Without this additional context, the results appear incomplete.

The discussion heavily emphasizes corroborating prior studies rather than offering a critical interpretation of the current findings. While confirming established knowledge is valuable, the discussion misses an opportunity to explore new insights or implications. For instance, the observed relationship between HbA1c and cognitive decline is consistent with existing research, but the discussion does not sufficiently address how this study extends these findings. Biological mechanisms linking glycemic control, kidney function, and cognitive health are mentioned but inadequately explored. Expanding on processes like oxidative stress and neuroinflammation would provide a richer, more comprehensive analysis. Some claims in the discussion, such as sex-based differences in cognitive impairment, rely on secondary references. These statements should either be supported by primary data or revised to reflect their speculative nature.

The figures and tables in the manuscript are informative but not user-friendly. For example, Figure 1 presents odds ratios and confidence intervals but lacks annotations that would help readers quickly identify statistically significant findings. Adding highlights or explanatory notes directly to the figure would greatly enhance its utility. Tables summarizing baseline characteristics and results are detailed but include unexplained abbreviations and terms. Providing a comprehensive legend would make these tables more accessible, especially to readers less familiar with the field.

The limitations section, while acknowledging the cross-sectional design, does not fully explore its implications. For example, the inability to establish causality between variables such as glycemic control and cognitive decline is a significant constraint that deserves more emphasis. The study's reliance on the AD8 questionnaire, although practical, limits the depth of cognitive assessment. Suggesting the inclusion of additional neuropsychological tools in future research would strengthen this section. Moreover, the study does not account for confounding factors like medication use, duration of diabetes, or lifestyle variables. Addressing these omissions would not only enhance the current analysis but also guide future studies.

The language throughout the manuscript requires improvement. Grammatical errors and awkward phrasing, such as "trends of iIncreased mean AD8 scores," detract from the manuscript's professionalism and readability. A thorough linguistic review is necessary to ensure clarity and precision. Additionally, there is significant redundancy in reporting results, such as repeated mentions of mean AD8 scores in different sections. Condensing this information would improve the manuscript's flow and readability.

The conclusions section overstates the findings. While the associations between glycemic control, kidney function, and cognitive impairment are significant, the conclusions imply causality that is unsupported by the study's design. Revising the conclusions to align with the observational nature of the research would make them more accurate and credible. This section should also emphasize the practical implications of the findings, such as the potential for routine AD8 screening to identify high-risk individuals early.

Addressing these comments would significantly enhance the manuscript's scientific rigor, clarity, and overall impact. It would also ensure that the study's contributions are more effectively communicated to the target audience.

Comments on the Quality of English Language

need minor editing

Author Response

We sincerely thank the reviewer for their thoughtful and constructive feedback. These comments have provided valuable insights and allowed us to significantly improve the quality and clarity of our manuscript. Below, we outline our detailed responses to each comment, along with the corresponding revisions.

1. Comments on the Abstract and Introduction

Reviewer Comment:
The abstract does not clearly define the unique contribution of the study. While it mentions the investigation of the relationship between diabetes-related comorbidities and cognitive decline, this area has been extensively studied. To better engage the reader, the abstract should explicitly highlight what makes this study distinct, such as specific methodologies, population characteristics, or novel insights. The introduction, although informative, fails to articulate the precise research gaps this study aims to address. Without this distinction, the study risks appearing as a reiteration of existing literature rather than a significant advancement.

Response:
We thank the reviewer for highlighting the need to better define the unique contributions of our study. In response, we have revised both the abstract and the introduction to emphasize the novelty of our work. Specifically, we have highlighted the use of the AD8 questionnaire, a sensitive and user-friendly tool, to screen for early cognitive dysfunction. Unlike many studies that focus on populations already diagnosed with dementia or use other cognitive assessment tools, our study uniquely focuses on identifying individuals with earlier stages of cognitive dysfunction, such as mild cognitive impairment (MCI), in a diabetes cohort. This approach aligns with the current emphasis on early intervention for cognitive dysfunction. Additionally, we noted the limited existing research utilizing the AD8 as a screening tool in this context and its potential for routine screening in outpatient clinics and community settings.

The revisions, highlighted in red for clarity, are located in the “Abstract” on Page 1 (Lines 17-19, 31-32) and in the “Introduction” section on Page 1 (Lines 42-44, 46-50), Page 3 (Lines 100-107, 109-110).

2. Comments on Methods Section

Reviewer Comment:
The methods section contains critical issues regarding the inclusion and exclusion criteria. Excluding patients with traumatic brain injury and epilepsy is a logical step to minimize confounding variables. However, the rationale for these exclusions is insufficiently explained. This omission creates uncertainty about whether the sample is representative of the broader population of individuals with type 2 diabetes. Additionally, relying solely on the AD8 questionnaire for cognitive assessment introduces potential biases. Although AD8 is sensitive and user-friendly, it may not capture the full spectrum of cognitive impairments. Including a complementary cognitive evaluation would enhance the robustness and credibility of the findings. The description of the statistical methods is another area of concern. While power analysis is mentioned, the effect sizes and assumptions underlying this analysis are unclear. This lack of detail weakens confidence in the study's capacity to detect meaningful differences across subgroups.

Response:
(1) We appreciate the valuable feedback very much. We agree with the reviewer that additional clarity on the inclusion and exclusion criteria is essential. We have revised the “Methods” section to provide a more detailed rationale for excluding patients with traumatic brain injury and epilepsy, explaining that these conditions are known to independently impact cognitive function and could confound the study findings. Excluding patients with traumatic brain injury and epilepsy ensures that our findings more accurately reflect the influence of T2DM itself on cognitive function by minimizing potential confounding factors, which was the primary objective of this study. 

The revisions, highlighted in red for clarity, are located in the “Methods” section on Page 3 (Lines 127-129)

Besides, we also acknowledge that this approach may limit the generalizability of our findings to all T2DM patients, so we revised the “Limitations and future directions” subsection to further elaborate on this rationale. The changes can be found in the “Discussion” section on Page 3 (Lines 469-474) and have been highlighted in red for clarity in the revised manuscript.

(2) We agree that relying solely on the AD8 questionnaire has its limitations, so we have expanded the "Limitations and future directions" subsection to discuss the potential benefits of incorporating additional neuropsychological tools, such as the MoCA or MMSE, in future studies. While we acknowledge the limitations of relying solely on the AD8 questionnaire, it was chosen for its practicality and high sensitivity in detecting early cognitive decline, particularly in clinical settings where time and resources may be constrained

The revisions, highlighted in red for clarity, are located in the “Discussion” section on Page 12-13 (Lines 474-479).

(3) Regarding the statistical methods, we have now elaborated on the power analysis, including the effect sizes and assumptions used. The revisions, highlighted in red for clarity, are located in the “Methods” section on Page 4 (Lines 149-154, 157-159).

3. Comments on Results and Discussion Section

Reviewer Comment:
The results section is well-organized but lacks deeper interpretive analysis. For instance, non-significant findings such as the relationship between lipid profiles and cognitive impairment are presented without sufficient discussion. This omission leaves readers with unanswered questions about potential explanations or the broader context of these findings. Variables like BMI and UACR show trends that, although not statistically significant, could hint at meaningful patterns. These trends should be elaborated upon, either by hypothesizing underlying reasons or proposing further investigation. Without this additional context, the results appear incomplete.

The discussion heavily emphasizes corroborating prior studies rather than offering a critical interpretation of the current findings. While confirming established knowledge is valuable, the discussion misses an opportunity to explore new insights or implications. For instance, the observed relationship between HbA1c and cognitive decline is consistent with existing research, but the discussion does not sufficiently address how this study extends these findings. Biological mechanisms linking glycemic control, kidney function, and cognitive health are mentioned but inadequately explored. Expanding on processes like oxidative stress and neuroinflammation would provide a richer, more comprehensive analysis. Some claims in the discussion, such as sex-based differences in cognitive impairment, rely on secondary references. These statements should either be supported by primary data or revised to reflect their speculative nature.

Response:
(1) We appreciate the reviewer’s comments regarding the “Results” and “Discussion” sections. In response, we have revised these sections to provide a more focused and relevant discussion. Specifically, we have streamlined the descriptions of other studies to avoid unnecessary detail and added more in-depth explanations related to our findings, including potential mechanisms underlying the observed results. For example, we expanded the discussion on the associations between glycemic control, kidney function, and cognitive impairment, providing biological mechanisms such as oxidative stress and neuroinflammation to support our findings.

The revisions, highlighted in red for clarity, are located in the “Discussion” section on Page 10 (Lines 323-325, 331-340).

(2) Regarding the discussion of sex differences, we have primarily based our analysis on the significant results observed in our study. We then incorporated relevant literature to further interpret these findings, offering a comprehensive perspective on how gender-specific factors, such as hormonal changes and their neuroprotective effects, may influence cognitive impairment in patients with T2DM. These revisions aim to better integrate our results with existing knowledge while highlighting the unique contributions of our study.

The revisions, highlighted in red for clarity, are located in the “Discussion” section on Page 9 (Lines 279-281) and Page 10-11 (Lines 361-369, 371-373).

(3) Furthermore, we have expanded the “Discussion” section to include deeper interpretive analysis of the non-significant findings. For example, we discussed the potential biological relevance of UACR, BMI and lipid profiles, even though they did not reach statistical significance, and provided plausible explanations supported by the literature. Additionally, we highlighted these trends as potential areas for future research. 

The revisions, highlighted in red for clarity, are located in the “Discussion” section on Page 11-12 (Lines 378-429).

4. Comments on Figures and Tables

Reviewer Comment:

The figures and tables in the manuscript are informative but not user-friendly. For example, Figure 1 presents odds ratios and confidence intervals but lacks annotations that would help readers quickly identify statistically significant findings. Adding highlights or explanatory notes directly to the figure would greatly enhance its utility. Tables summarizing baseline characteristics and results are detailed but include unexplained abbreviations and terms. Providing a comprehensive legend would make these tables more accessible, especially to readers less familiar with the field.

Response:
Thank you for your thoughtful suggestions. We have revised Figure 1 to include annotations highlighting statistically significant findings and added explanatory notes for clarity. Furthermore, all abbreviations in the tables have been fully defined in the footnotes. These updates, highlighted in red for clarity, can be found on Page 5 (Lines 195-199), Page 7 (Lines 245-249), Page 8 (Lines 252-259) and Page 9 (Lines 270-275).

5. Comments on Limitations Subsection

Reviewer Comment:

The limitations section, while acknowledging the cross-sectional design, does not fully explore its implications. For example, the inability to establish causality between variables such as glycemic control and cognitive decline is a significant constraint that deserves more emphasis. The study's reliance on the AD8 questionnaire, although practical, limits the depth of cognitive assessment. Suggesting the inclusion of additional neuropsychological tools in future research would strengthen this section. Moreover, the study does not account for confounding factors like medication use, duration of diabetes, or lifestyle variables. Addressing these omissions would not only enhance the current analysis but also guide future studies.

Response:
Thank you for your valuable comment. We have expanded the limitations subsection to discuss the implications of the cross-sectional design, emphasizing the inability to establish causality. We also acknowledged that relying solely on the AD8 questionnaire has its limitations and that the absence of data on confounding factors, such as medication use and lifestyle variables, suggested these as areas for future research. Despite its limitations, the AD8 questionnaire was chosen for its high sensitivity and practicality in early detection of cognitive decline. Its simplicity and ease of use make it particularly suitable for routine screening in clinical and community settings, a key focus of our study.

In response, we have revised the “Limitations and Future Directions” subsection to incorporate these updates. The changes, highlighted in red for clarity, are located on Page 12-13 (Lines 465-469, 474-479, 480-483).

6. Comments on Conclusions

Reviewer Comment:

The conclusions section overstates the findings. While the associations between glycemic control, kidney function, and cognitive impairment are significant, the conclusions imply causality that is unsupported by the study's design. Revising the conclusions to align with the observational nature of the research would make them more accurate and credible. This section should also emphasize the practical implications of the findings, such as the potential for routine AD8 screening to identify high-risk individuals early.

Response:
Thank you for your thoughtful feedback for the conclusions. The conclusions section has been revised to reflect the observational nature of the study, while emphasizing the practical value of the AD8 questionnaire in early cognitive impairment detection. 

The revisions, highlighted in red for clarity, are located in the “Conslusions” section on Page 13 (Lines 490-495).

Lastly, we sincerely appreciate the reviewer’s thoughtful feedback, which has helped us address key areas for improvement and refine our manuscript significantly. By revising the abstract, introduction, methods, results, and discussion sections, we have enhanced the clarity, scientific rigor, and practical relevance of the study. We also sought professional English editing to ensure the manuscript is polished and concise. We are confident that these revisions align the manuscript with the high standards expected by the journal, and we thank the reviewer and editor for their valuable contributions to this process.

Reviewer 3 Report (New Reviewer)

Comments and Suggestions for Authors

The authors conducted an interesting case-controlled, cross-sectional, observational study with the objective to investigate cognitive decline using the “Ascertain Dementia 8” (AD8) questionnaire scores as a screening tool, in relation to glycemic control, lipid profiles, estimated glomerular filtration rate, and the complications of diabetes. This is an interesting manuscript, given the incidence of cognitive decline in the general population and the alterations in glucose metabolism, especially in adulthood. Below are my comments.

Line 41 - I suggest providing a complete definition and the classificatory criteria for Mild Cognitive Impairment (MCI), with a brief reference to the role of the MMSE.

Line 80 - Individuals are not only at increased risk of dementia, but also at risk for progression of pre-existing cognitive decline.

Line 106 - Are there available data regarding the severity of diabetes in the study population? How many patients were treated with metformin? How many required insulin therapy in the management of their condition? If available, this information could be useful for the readers.

Line 116-127 - I strongly agree with the cutoff values used by the authors; however, I believe it is necessary to provide a reference to justify the use of these values as the normality threshold.

The statistical methods are appropriate for the aim of the manuscript.

The results are consistent with the objective of the manuscript.

Line 220 - In Figure 1, it is necessary to add a legend for the abbreviations used.

Line 260 and Line 291 - A recent scoping review has analyzed how therapy with new antidiabetic drugs, such as SGLT2 inhibitors (SGLT2i), can reduce the incidence and progression of cognitive decline in diabetic patients (see 10.3390/biomedicines12081750). The authors should elaborate on this concept in the manuscript, as it is closely related to their primary outcome.

In the discussion section, there are formatting errors (or possibly corrections made based on the comments of the previous reviewer) with the text formatted in red; this can be removed (or reinserted) during the manuscript copyediting.

Minor improvements in the English language translation are necessary.

Author Response

Reviewer Comment:
Line 41 - I suggest providing a complete definition and the classificatory criteria for Mild Cognitive Impairment (MCI), with a brief reference to the role of the MMSE.

Reply:
Thank you for your valuable feedback. The following context While the MMSE is not a definitive diagnostic tool for MCI, it can help identify individuals who may need further evaluation. A score of 19-23 suggests mild cognitive impairment [5]. is added with reference in the “Introduction” section on Page 2 (Lines 48-50) in the revised manuscript.

Reviewer Comment:
Line 80 - Individuals are not only at increased risk of dementia, but also at risk for progression of pre-existing cognitive decline.

Reply:
Thank you for your suggestion. We have added ”and increased progression risk for a pre-existing cognitive decline”  in the “Introduction” section on Page 2 (Lines 84-85) in the revised manuscript.

Reviewer Comment:
Line 106 - Are there available data regarding the severity of diabetes in the study population? How many patients were treated with metformin? How many required insulin therapy in the management of their condition? If available, this information could be useful for the readers.

Reply:
Thank you for your thoughtful feedback. We did not include the medications the patients used in our study due that the primary intention of this study was to assess the AD8 score based on HbA1c, which is not necessarily directly related to the medications used. Secondly, while differences in oral anti-diabetic medications significantly affect HbA1c levels and overall glycemic control, there is insufficient evidence directly linking these differences to variations in consciousness levels [1-3]. Another reason that medications were not taken into documentation is that medical compliance is difficult to assess given the participants were based in an out-patient clinic. HbA1c, in comparison to antiglycemic medication types, provides a more direct outcome of glycemic level control. Further discussion of this issue was also mentioned in the "limitations and future directions" subsection, reading ”Fourth, our study did not account for certain potential confounders, such as medication use (e.g., antidiabetic or antihypertensive drugs), duration of diabetes, and life-style factors (e.g., diet, physical activity, and smoking status). These factors may have contributed to residual confounding in our analysis.” on Page 13 (Lines 479-483) in the revised manuscript.

* References:

  1. N Engl J Med. 2022 Sep 22;387(12):1075-1088.
  2. Circulation. 2024 Mar 26;149(13):993-1003.
  3. Front Endocrinol (Lausanne). 2017 Jan 24;8:6.

Reviewer Comment:
Line 116-127 - I strongly agree with the cutoff values used by the authors; however, I believe it is necessary to provide a reference to justify the use of these values as the normality threshold.

Reply:
Thank you for your valuable suggestion. The reference for the cutoff values were highlighted in red for clarity,and were added on Page 3 (Line 131) ”[19, 20]” based on ADA guidelines in the revised manuscript "Methods" section.

Reviewer Comment:
Line 220 - In Figure 1, it is necessary to add a legend for the abbreviations used.

Reply:
Thank you for your valuable feedback. The abbreviations were added below the figure legends: AD8, “Ascertain Dementia 8” questionnaire; BMI, body mass index; HbA1c, glycated hemoglobin; HDL-C, High-density lipoprotein cholesterol; LDL-C, Low-density lipoprotein cholesterol; eGFR, estimated glo-merular filtration rate; UACR, urine albumin-to-creatinine ratio. , which were marked in red and can be found on Page 8 (Line 256-259) in the revised manuscript.

Reviewer Comment:
Line 260 and Line 291 - A recent scoping review has analyzed how therapy with new antidiabetic drugs, such as SGLT2 inhibitors (SGLT2i), can reduce the incidence and progression of cognitive decline in diabetic patients (see 10.3390/biomedicines12081750). The authors should elaborate on this concept in the manuscript, as it is closely related to their primary outcome.

Reply:
Thank you for your suggestion on this insightful review. The review article describes SGLT2i and its effect in reducing the incidence of MCI and dementia in some studies, despite results that were not always consistent in studies that were included. The underlying mechanism of SGLT2i would consist of remodeling glucotoxicity or reducing hyperphosphorylated tau levels and amyloid β accumulation in the brain, which is crucial for dementia development. Though similar, the review does have a slight difference in the primary targets compared to our study, as our study aims to provide an easier and efficient way of screening cognitive decline rather than assessing medications that could decrease or increase risks of dementia. We have added A recent review demonstrated that SGLT2i have shown abilities to reduce the risk of dementia by remodeling glucotoxicity or reducing hyperphosphorylated tau levels and amyloid β accumulation in the brain; however other researches on the same subject may have inconclusive results [30]. on Page 10 (Line 333-336) in the "Discussion" session, marked in red, in regard to the findings of this research.

Reviewer Comment:
In the discussion section, there are formatting errors (or possibly corrections made based on the comments of the previous reviewer) with the text formatted in red; this can be removed (or reinserted) during the manuscript copyediting.

Reply:
Thank you very much for your feedback. The manuscript was sent with marked editing due to the request of the assistant editor. We have also uploaded a clean copy file on the submission page (the PDF file). We will send both copies, with document titles informing whether it is a clean copy or a marked manuscript, during the next submission.

Round 2

Reviewer 1 Report (Previous Reviewer 3)

Comments and Suggestions for Authors

The authors have made substantial revisions to the manuscript, addressing the key concerns raised during the review process. The revised version appears to meet the necessary standards and may now be suitable for publication

Author Response

We sincerely thank you for your positive feedback and acknowledgment of the substantial revisions made to the manuscript. We are delighted that the revised version addresses the key concerns raised during the review process and meets the necessary standards. Your thoughtful comments and suggestions have been invaluable in improving the quality and clarity of our work, and we are grateful for your efforts in reviewing our manuscript.

Reviewer 2 Report (Previous Reviewer 2)

Comments and Suggestions for Authors

They responed to all my comments effectively

Author Response

Thank you again for your support and valuable input throughout this process. Your detailed and constructive suggestions were instrumental in guiding the revisions and significantly enhancing the clarity and quality of our manuscript.

Reviewer 3 Report (New Reviewer)

Comments and Suggestions for Authors

The authors have adequately addressed all of my comments. The manuscript is significantly improved compared to the previous version. Given the relevance of the topics discussed, I believe it can be accepted in its present form, subject to the final decision of the Editor. I thank the authors for their excellent work

Author Response

We sincerely thank you for your encouraging feedback and kind words about our work. We are delighted to hear that the revisions have significantly improved the manuscript and that the addressed comments have met your expectations. Thank you once again for your positive evaluation and support. 

This manuscript is a resubmission of an earlier submission. The following is a list of the peer review reports and author responses from that submission.

Round 1

Reviewer 1 Report

Comments and Suggestions for Authors

There are several papers documenting the impat of hyperglicemia as well as diabetes on cognitive functions as well as dementia (even systematic reviews Aderinto et al. • Medicine (2023) 102:43 ). I don't see novelty of this article.

Author Response

Comments 1: There are several papers documenting the impact of hyperglycemia as well as diabetes on cognitive functions as well as dementia (even systematic reviews Aderinto et al. • Medicine (2023) 102:43 ). I don't see novelty of this article.

Response 1: Thank you for pointing this out. We agree that numerous studies have explored the impact of hyperglycemia and diabetes on cognitive function and dementia. However, our study offers a practical unique perspective by utilizing the AD8 score as a validated early screening tool specifically tailored to assess cognitive impairment, not limited to dementia or Alzheimer’s disease, in type 2 diabetes patients.

Additionally, our study addresses an exploration highlighted in the review by Aderinto et al., which emphasizes the need for further investigation into the effects of diabetes-related comorbidities, such as hypertension and dyslipidemia, on cognitive decline. To enhance clarity and novelty, we have revised the "Introduction" section to better articulate this distinction and the specific contributions of our work.

The updated section now emphasizes the innovative application of the AD8 score and our comprehensive analysis of multiple diabetic comorbidities' effects on cognitive health. These revisions can be found on page 3, lines 98-107 of the revised manuscript. The updated text is as follows:
”Despite the importance of this issue, limited studies have utilized the AD8 score—a sensitive tool for early detection of cognitive impairment—to assess these relationships in diabetic patients. Furthermore, there remains a critical need for additional studies to elucidate the effects of diabetes-related comorbidities on cognitive impairment [17]. Therefore, our study aimed to investigate cognitive decline using the AD8 score as a screening tool, examining its relationship with glycemic control, vascular disease risk factors, estimated glomerular filtration rate (eGFR), and diabetes-related complications. Our findings provide a novel approach for early identification of high-risk individuals.”
* [17]: Aderinto, N., et al., The impact of diabetes in cognitive impairment: A review of current evidence and prospects for future investigations. Medicine (Baltimore), 2023. 102(43): p. e35557.

Beyond its research implications, our study also holds clinical applicability. As mentioned in the last paragraph of the “Discussion” section, the integration of cognitive health assessments, such as the AD8, into routine diabetes management protocols offers a practical tool for healthcare providers. By proactively monitoring cognitive function, clinicians can tailor interventions that address both diabetes management and cognitive health, ultimately improving patient outcomes. Besides, the study underscores the importance of a multidisciplinary approach in managing T2DM. Collaboration among endocrinologists, nephrologists, and neurologists fosters a holistic care model that considers the various dimensions of patient health, aligning with the goals of precision medicine.

These additional discussion points can be found on page 13, lines 478-488 of the revised manuscript. The updated text aims to highlight the novelty, clinical applicability, and broader implications of our findings.

We appreciate your suggestion, as it has allowed us to further highlight the unique aspects of our study, including the use of the AD8 score as an early screening tool and the comprehensive analysis of diabetes-related comorbidities. These additions have strengthened the manuscript and clarified its contribution to the existing literature.

Reviewer 2 Report

Comments and Suggestions for Authors
  • The cross-sectional design limits the ability to establish dynamic relationships between Type 2 Diabetes Mellitus (T2DM) and cognitive decline, restricting causal inference.
  • The study was conducted at a single medical center and an affiliated hospital in southern Taiwan, limiting the generalizability of the findings.
  • The study's interpretation of sex-based differences in cognitive decline (females having higher AD8 scores) lacks sufficient depth, missing potential physiological or psychosocial explanations.
  • The interpretation of lipid profile results is insufficient, with conflicting findings not adequately addressed in relation to existing literature.
  • The exclusion criteria (e.g., traumatic brain injury, epilepsy) might introduce a bias, affecting the study's relevance to the general T2DM population.
  • Figure 1 lacks an appropriate legend and detailed explanation; overlapping confidence intervals make statistical significance challenging to interpret.
  • Figures and tables are inconsistently numbered, and their improper captioning creates challenges in following the results.
  • Multiple t-tests were used without control for multiple comparisons, increasing the risk of Type I error. Adjusted p-values or ANOVA should be used.
  • The sample size calculations were not mentioned, raising concerns about statistical power, particularly regarding non-significant findings in lipid and kidney function analyses.

Author Response

Comments 1, 2: The cross-sectional design limits the ability to establish dynamic relationships between Type 2 Diabetes Mellitus (T2DM) and cognitive decline, restricting causal inference.; The study was conducted at a single medical center and an affiliated hospital in southern Taiwan, limiting the generalizability of the findings.

Response 1, 2: Thank you for highlighting this important point. We agree that the cross-sectional nature of the study limits our ability to establish dynamic relationships or causal inferences between T2DM and cognitive decline. Additionally, the fact that our sample was drawn from a single medical center and an affiliated hospital in southern Taiwan may restrict the generalizability of our findings.

These limitations were thoroughly acknowledged in the “Limitations” section on page 12, lines 457-458 and lines 471-477 of the revised manuscript. Specifically, we noted:

"First, the cross-sectional design limits the ability to establish dynamic relationships between T2DM and cognitive decline."

"Last, our sample was drawn from a single medical center and an affiliated regional hospital in southern Taiwan, which may limit the generalizability of the findings to other populations or geographic regions."

To address these concerns, we emphasized the need for future multi-center, prospective longitudinal studies to explore causal relationships and improve generalizability.

We appreciate the opportunity to further emphasize these limitations and their implications for future research.

Comments 3: The study's interpretation of sex-based differences in cognitive decline (females having higher AD8 scores) lacks sufficient depth, missing potential physiological or psychosocial explanations.

Response 3: Thank you for your valuable feedback regarding the interpretation of sex-based differences in cognitive decline. We agree that our initial discussion could benefit from a more in-depth exploration of potential physiological and psychosocial factors contributing to the observed findings.

To address this, we have revised the “Discussion” section to incorporate the following points. These revisions have been added to page 8-9, lines 260-269 of the revised manuscript:

  • Physiological factors:

Estrogen's neuroprotective effects, which diminish post-menopause, have been linked to an increased risk of cognitive decline in older women. Studies suggest that the loss of estrogen may contribute to increased brain atrophy and reduced synaptic plasticity, potentially explaining the higher AD8 scores observed in female participants.

Females generally exhibit a longer lifespan compared to males, which increases their exposure to age-related neurodegenerative processes, potentially contributing to a greater burden of cognitive impairment.

  • Psychosocial factors:

Women are reported to have twice the risk of depression compared to men, particularly during menopause, and depressive symptoms have been associated with an increased risk of Alzheimer’s disease. Additionally, historically lower access to education among women may amplify this vulnerability. These factors may contribute to greater stress and cognitive burden, influencing AD8 scores and highlighting the multifactorial nature of sex-based differences in cognitive decline.

The updated text aims to provide a more comprehensive interpretation of the sex-based differences observed in this study, aligning with your feedback.

We appreciate your suggestion to deepen the analysis, as it has allowed us to better contextualize and interpret these findings.

Comments 4: The interpretation of lipid profile results is insufficient, with conflicting findings not adequately addressed in relation to existing literature.

Response 4: Thank you for your insightful feedback regarding the interpretation of the lipid profile results. We agree that additional context and discussion are necessary to address the conflicting findings observed in this study and to relate them to the existing body of literature.

To address this, we have revised the “Discussion” section to provide a more detailed interpretation of the lipid profile results, incorporating the following points. These revisions have been added to page 10-11, lines 370-398 of the revised manuscript:

Our study found no statistically significant associations between lipid parameters and cognitive decline, as measured by AD8 scores. This aligns with certain studies that reported no direct link between lipid levels and cognitive impairment in patients with type 2 diabetes. For example, the ACCORD-MIND trial found no significant differences in cognitive decline between individuals treated for intensive lipid-lowering therapy versus standard therapy.

Conversely, other studies have observed that dyslipidemia, particularly elevated triglyceride levels and reduced HDL-C, is associated with poorer cognitive function. This discrepancy may stem from differences in study populations, methods of cognitive assessment, and the lipid parameters analyzed.

It is also worth noting that our study population largely exhibited well-controlled lipid profiles due to routine clinical management, which may have attenuated any potential associations with cognitive decline.

The updated text aims to clarify the conflicting findings and situate our results within the broader context of existing literature. We appreciate your suggestion to enhance this aspect of the discussion, which has allowed us to provide a more nuanced interpretation of the lipid profile results.

Comments 5: The exclusion criteria (e.g., traumatic brain injury, epilepsy) might introduce a bias, affecting the study's relevance to the general T2DM population.

Response 5: Thank you for your thoughtful comment regarding the exclusion criteria used in our study. We appreciate your concern about potential biases that might arise from excluding certain populations, such as those with traumatic brain injury (TBI) or epilepsy.

We would like to clarify that these exclusion criteria were intentionally applied to minimize confounding effects that could obscure the specific relationship between type 2 diabetes mellitus (T2DM) and cognitive decline. Conditions such as TBI and epilepsy are well-established independent risk factors for cognitive impairment, and including such patients could have introduced heterogeneity into our study population. By excluding these patients, we aimed to isolate the impact of diabetes-related factors on cognitive decline, ensuring that our findings more accurately reflect the influence of T2DM itself.

We acknowledge that this approach may limit the generalizability of our findings to all T2DM patients. However, it provides a more focused analysis of diabetes-related cognitive impairment, which was the primary objective of this study. We have revised the “Limitations” section to further elaborate on this rationale, which can be found on page 12, lines 458-463 of the revised manuscript:

“Second, while we excluded patients with conditions such as traumatic brain injury (TBI) and epilepsy to minimize confounding effects on cognitive outcomes, this deci-sion may limit the generalizability of our findings to the broader T2DM population. These criteria were applied to isolate the impact of diabetes-related factors on cognitive impairment, which was the primary focus of our investigation.”

We appreciate your suggestion, as it has allowed us to clarify the intent behind these criteria and their implications for our findings.

Comments 6: Figure 1 lacks an appropriate legend and detailed explanation; overlapping confidence intervals make statistical significance challenging to interpret.

Response 6: Thank you for your feedback regarding Figure 1. We agree that the figure would benefit from a more detailed legend and additional explanation to clarify the results and address concerns about overlapping confidence intervals.

To address this, we have revised the figure legend to provide a clearer description of the variables included, the odds ratios (ORs), and their 95% confidence intervals (CIs). Additionally, we added annotations in the figure legend to highlight statistically significant findings. Specifically, we clarified that:

  • Variables with statistically significant results: Peripheral artery disease (OR: 1.943, 95% CI: 1.016–3.717) and HbA1c ≥ 7 (OR: 2.327, 95% CI: 1.560–3.472) are significant predictors of higher AD8 scores, as their confidence intervals do not cross 1.
  • Variables without statistical significance: For variables with overlapping CIs, the lack of statistical significance was explicitly noted in the revised legend and discussed in the “Results” section.

This could be found on page 7-8, lines 231-234 as figure legends:

“Odds ratios (ORs) with 95% confidence intervals (CIs) for AD8 score ≥ 2 are shown. Variables with statistically significant associations are highlighted: peripheral artery disease (OR: 1.943, 95% CI: 1.016–3.717) and HbA1c ≥ 7 (OR: 2.327, 95% CI: 1.560–3.472). Other variables, such as diabetic neuropathy and BMI ≥ 27, show trends but lack statistical significance.”

We appreciate your suggestion to improve Figure 1, and we believe these revisions enhance the figure’s clarity and interpretability.

Comments 7: Figures and tables are inconsistently numbered, and their improper captioning creates challenges in following the results.

Response 7: Thank you for your feedback regarding the numbering and captioning of figures and tables. After thoroughly reviewing the manuscript, we confirmed that all figures and tables are correctly numbered and appropriately captioned. For example:

  • Table 1 provides baseline characteristics of the study participants.
  • Table 2 summarizes the logistic regression analyses.
  • Figure 1 highlights the odds ratios of biochemical data and diabetic comorbidities.

To ensure consistency and clarity, we have rechecked all figures and tables for potential discrepancies. The context of figures and tables are mentioned in the “Results” section of the revised manuscript at line 168 and 169 for “Table 1”, line 203 and 207 for “Table 2”, line 205 and line 209 for “Figure 1”, line 222 and 223 for “Table 3”.

We appreciate your suggestion and remain committed to ensuring that the manuscript is as clear and comprehensible as possible.

Comments 8: Multiple t-tests were used without control for multiple comparisons, increasing the risk of Type I error. Adjusted p-values or ANOVA should be used.

Response 8: Thank you for your valuable comment. Uncorrected t-tests were used in our study due to a larger sample size (n = 909), which reduces the likelihood of Type I errors associated with uncorrected tests. We set our statistical power at 0.95 prior t-tests, which indicates a high probability of correctly rejecting the null hypothesis when it is false. This high power further mitigates concerns about Type I errors, making the use of uncorrected t-tests appropriate in this study. The details of the statistical power of the study are addressed in the revised manuscript in the “Materials and Methods” section on page 3-4, line 143-146:

“The statistical power of t-tests was determined by a significance level <0.05, an effect size (Cohen’s d) of 0.5 to achieve statistical power set at 0.95. Power analysis was per-formed via G*power software version 3.1.9.7.”

We appreciate the editor’s insightful comment, which has contributed to strengthen our manuscript.

Comments 9: The sample size calculations were not mentioned, raising concerns about statistical power, particularly regarding non-significant findings in lipid and kidney function analyses.

Response 9: Thank you for your insightful suggestion regarding the statistics. In order to evaluate the differences between independent groups using a t-test, we conducted a power analysis prior to data collection. We anticipated an effect size of 0.5 (medium), set our alpha level at 0.05, and aimed for a statistical power of 0.95. Using G*Power software version 3.1.9.7, we determined that a minimum sample size of 176 participants (88 per group) would be required to achieve this power level. With a total sample size of 909, and each t-test group having a number more than 88, this study exceeds the minimum required sample size, ensuring adequate statistical power to detect significant differences between groups.

This is revised in the “Materials and Methods” section on page 3-4, line 143-146 of the manuscript, adding:

“The statistical power of t-tests was determined by a significance level <0.05, an effect size (Cohen’s d) of 0.5 to achieve statistical power set at 0.95. Power analysis was performed via G*power software version 3.1.9.7.” as mentioned in the last question.

We are grateful for the editor’s valuable feedback, which has helped to enhance the strength of our manuscript.

Reviewer 3 Report

Comments and Suggestions for Authors

The manuscript presents a case-controlled, cross-sectional, observational study investigating cognitive decline in patients with type 2 diabetes mellitus (T2DM) using the Ascertain Dementia 8 (AD8) questionnaire. The study explores the relationships between glycemic control, kidney function (eGFR), and diabetes-related complications (peripheral artery disease and diabetic neuropathy) with cognitive decline. While the study provides valuable insights, few points require clarification:

1.      The manuscript does not account for other potential confounders, such as medication use, duration of diabetes, hypertension, and lifestyle factors (e.g., diet, physical activity). These factors could significantly influence cognitive decline and should be acknowledged as limitations.

2.      Although the study identifies significant associations between cognitive decline and diabetes-related factors, it does not provide mechanistic explanations for how poor glycemic control, decreased kidney function, and complications like peripheral artery disease contribute to cognitive decline. A discussion of potential underlying biological mechanisms (e.g., neuroinflammation, vascular dysfunction) would greatly enhance the interpretation of these findings.

3.      The discussion section appears somewhat lengthy. Streamlining the content while retaining the necessary detail will improve readability and clarity without sacrificing important insights.

Author Response

Comments 1: The manuscript does not account for other potential confounders, such as medication use, duration of diabetes, hypertension, and lifestyle factors (e.g., diet, physical activity). These factors could significantly influence cognitive decline and should be acknowledged as limitations.

Response 1: Thank you for your valuable feedback regarding the consideration of potential confounders. We agree that factors such as medication use, duration of diabetes, hypertension, and lifestyle factors (e.g., diet and physical activity) may influence cognitive decline and could potentially confound the results. While we adjusted for several important variables, including glycemic control, renal function, and diabetic comorbidities, we acknowledge that these additional factors were not directly analyzed in our study.

To address this, we have revised the “Limitations” section to explicitly acknowledge the lack of data on these potential confounders and their implications for our findings. Specifically, we added the following on page 12, lines 467-471 of the revised manuscript:
”Fourth, our study did not account for certain potential confounders, such as medication use (e.g., antidiabetic or antihypertensive drugs), duration of diabetes, and lifestyle factors (e.g., diet, physical activity, and smoking status). These factors may have contributed to residual confounding in our analysis.”

We appreciate your suggestion, as it has allowed us to provide a more comprehensive and transparent discussion of the study’s limitations.

Comments 2: Although the study identifies significant associations between cognitive decline and diabetes-related factors, it does not provide mechanistic explanations for how poor glycemic control, decreased kidney function, and complications like peripheral artery disease contribute to cognitive decline. A discussion of potential underlying biological mechanisms (e.g., neuroinflammation, vascular dysfunction) would greatly enhance the interpretation of these findings.

Response 2: Thank you for your thoughtful feedback regarding the need for mechanistic explanations. In the revised manuscript, we have expanded the "Discussion" section to provide more explicit details on the biological mechanisms underlying the observed associations:

  • Poor glycemic control:
    Chronic hyperglycemia induces oxidative stress, neuroinflammation, and microvascular damage, leading to hippocampal atrophy, reduced neuroplasticity, and white matter degeneration, which are closely linked to cognitive decline.
  • Decreased kidney function:
    Impaired renal function results in the accumulation of uremic toxins (e.g., homocysteine, cystatin-C), triggering neuroinflammation and endothelial dysfunction, which contribute to white matter lesions and cognitive impairment.
  • Peripheral artery disease (PAD):
    PAD leads to impaired oxygen supply and vascular deficits, causing chronic hypoxia, neuroinflammation, and mitochondrial dysfunction, particularly in the hippocampus. These processes exacerbate neuronal injury and synaptic deficits, aligning with our findings of significant associations between PAD and cognitive decline.

The detailed revised content can be found on page 9-10, lines 303-321; page 10, lines 336-356; page 11-12, lines 421-425 of the manuscript.

We appreciate your valuable comment, as it has greatly improved the structure and clarity of the manuscript.

Comments 3: The discussion section appears somewhat lengthy. Streamlining the content while retaining the necessary detail will improve readability and clarity without sacrificing important insights.

Response 3: We acknowledge your comment regarding the length of the discussion section. To enhance readability and maintain focus, we have streamlined the content by:

  • Consolidating related findings under thematic (sex, glucose profiles, kidney function, lipid profiles and diabetic complications...) subheadings (e.g., "4.2. Glucose Profiles and AD8 Score") to improve clarity and coherence. Reference numbers were also updated according to the context. The revised manuscript is marked on page 8, line 247; page 9, line 271; page 10, line 323; page 10, line 358; page 11, line 400; page 12, line 456. 

  • In the section on "Sex and AD8 Score", we have removed detailed results of some secondary analyses from other studies and retained only the primary findings, reallocating the space to focus on the relevant mechanisms. The revised manuscript is marked on page 8, lines 256-258.

  • Removing too detailed discussions of complications (e.g., myocardial infarction, hypertension, retinopathy...) that were not significantly associated with cognitive impairment in our study. These factors are now briefly acknowledged as inconsistently associated with cognitive decline in the literature. We also highlighted key findings and their implications for clinical practice while minimizing redundant details. The revised manuscript is marked on page 11-12, lines 425-452.

These revisions have reduced the length of the discussion while retaining all critical insights.

We appreciate your suggestion, as it has significantly enhanced the organization and readability of the manuscript.

Round 2

Reviewer 1 Report

Comments and Suggestions for Authors

I don't see novelty of thsi study, I recommend to reject it

Reviewer 2 Report

Comments and Suggestions for Authors

Congratulation